# A UNIFIED PATHS PERSPECTIVE FOR PRUNING AT INITIALIZATION

## ABSTRACT

A number of recent approaches have been proposed for pruning neural network parameters at initialization with the goal of reducing the size and computational burden of models while minimally affecting their training dynamics and generalization performance. While each of these approaches have some amount of well-founded motivation, a rigorous analysis of the effect of these pruning methods on network training dynamics and their formal relationship to each other has thus far received little attention. Leveraging recent theoretical approximations provided by the Neural Tangent Kernel, we unify a number of popular approaches for pruning at initialization under a single path-centric framework. We introduce the Path Kernel as the data-independent factor in a decomposition of the Neural Tangent Kernel and show the global structure of the Path Kernel can be computed efficiently. This Path Kernel decomposition separates the architectural effects from the data-dependent effects within the Neural Tangent Kernel, providing a means to predict the convergence dynamics of a network from its architecture alone. We analyze the use of this structure in approximating training and generalization performance of networks in the absence of data across a number of initialization pruning approaches. Observing the relationship between input data and paths and the relationship between the Path Kernel and its natural norm, we additionally propose two augmentations of the SynFlow algorithm for pruning at initialization.

## 1    INTRODUCTION

A wealth of recent work has been dedicated to characterizing the training dynamics and generalization bounds of neural networks under a linearized approximation of the network depending on its parameters at initialization (Jacot et al., 2018; Arora et al., 2019; Lee et al., 2019a; Woodworth et al., 2020). This approach makes use of the Neural Tangent Kernel, and under infinite width assumptions, the training dynamics of gradient descent over the network become analytically tractable. In this paper, we make use of the Neural Tangent Kernel theory with the goal of approximating the effects of various initialization pruning methods on the resulting training dynamics and performance of the network. Focusing on networks with homogeneous activation functions (ReLU, Leaky-ReLU, Linear), we introduce a novel decomposition of the Neural Tangent Kernel which separates the effects of network architecture from effects due to the data on the training dynamics of the network. We find the data-independent factor of the Neural Tangent Kernel to have a particularly nice structure as a symmetric matrix representing the covariance of path values in the network which we term the Path Kernel. We subsequently show that the Path Kernel offers a data-independent approximation of the network's convergence dynamics during training.

To validate the empirical benefits of this theoretical approach, we turn to the problem of pruning at initialization. While the problem of optimally pruning deep networks is nearly as old as deep networks themselves (Reed, 1993), interest in this problem has experienced a revival in recent years. This revival is likely a product of a number of underlying factors, but much of the recent interest could be ascribed to the Lottery Ticket Hypothesis (Frankle & Carbin, 2018) which states that sparse, trainable networks–which achieve task performance that matches or exceeds those of its dense counterparts–can be found *at initialization*. The Lottery Ticket Hypothesis implies that the over-parameterization of neural networks is incidental in finding a trainable solution, the topology of which often exists at initialization. However, finding these lottery ticket networks currently requires

some amount of iterative re-training of the network at increasing levels of sparsity which is inefficient and difficult to analyze theoretically.

The resurgence of interest in optimal pruning has spurred the development of a number of recent approaches for pruning deep neural networks at initialization (Lee et al., 2019b; Liu & Zenke, 2020; Wang et al., 2020; Tanaka et al., 2020) in supervised, semi-supervised, and unsupervised settings, borrowing theoretical motivation from linearized training dynamics (Jacot et al., 2018), mean-field isometry (Saxe et al., 2013), and saliency (Dhamdhere et al., 2019). While each of these methods have their own theoretical motivations, little work has been dedicated to formally describing the effect of these pruning methods on the expected performance of the pruned network. Also, the diversity in theoretical motivations that give rise to these pruning methods makes it difficult to observe their similarities.

In this paper, we observe that a number of initialization pruning approaches are implicitly dependent on the path covariance structure captured by the Path Kernel which, in turn, affects the network's training dynamics. We show that we can approximate these training dynamics in general, and our approximation results for a number of initialization pruning approaches suggests that it is possible to estimate, prior to training, the efficacy of a particular initialization pruning approach on a given architecture by investigating the eigenstructure of its Path Kernel. Motivated by our theoretical results and the unification of a number of initialization pruning methods in this Path Kernel framework, we investigate the close relationship between the SynFlow (Tanaka et al., 2020) pruning approach and our path decomposition. This leads to our suggestion of two new initialization pruning approaches which we predict to perform well under various assumptions on the stability of the Path Kernel and the input distribution of the data. We then validate these predictions empirically by comparing the performance of these pruning approaches across a number of network architectures.

The insights on initialization pruning provided by the Path Kernel decomposition are only one of a number of potential application domains which could benefit from this path-centric framework. Importantly, the coviariance structure over paths encoded by the Path Kernel is general and may be computed at any point in time, not just at initialization. We anticipate that this representation will provide insight into other application areas like model interpretation, model comparison, or transfer learning across domains.

The sections of the paper proceed as follows. We start with a brief introduction to the Neural Tangent Kernel in Section 2 before introducing the Path Kernel decomposition in Section 3 and its relationship to approximations of network convergence properties. In Section 4, we reformulate in this path framework three popular initialization pruning approaches and introduce two additional initialization pruning approaches inspired by this path decomposition. We validate these convergence approximations and the behavior of these pruning approaches in Section 5 and conclude with a discussion of the results and opportunities for future work.

## 2   THE NEURAL TANGENT KERNEL

Recent work by Jacot et al. (2018) has shown that the exact dynamics of infinite-width network outputs through gradient descent training corresponds to kernel gradient descent in function space with respect the **Neural Tangent Kernel**. More formally, for a neural network $f$ parameterized by $\boldsymbol{\theta}$ and loss function $\ell : \mathbb{R}^K \times \mathbb{R}^K \to \mathbb{R}$, let $\mathcal{L} = \sum_{(\boldsymbol{x} \in \boldsymbol{\mathcal{X}}, \boldsymbol{y} \in \boldsymbol{\mathcal{Y}})} \ell(f_t(\boldsymbol{x}, \theta), \boldsymbol{y})$ denote the empirical loss function. Here, $\boldsymbol{\mathcal{X}}$ is the training set, $\boldsymbol{\mathcal{Y}}$ is the associated set of class labels. For multiple inputs, denote $\boldsymbol{f}(\boldsymbol{\mathcal{X}}, \boldsymbol{\theta}) \in \mathbb{R}^{NK}$ the outputs of the network where $K$ is the output dimension and $N$ is the number of training examples. In continuous-time gradient descent, the evolution of parameters and outputs can be expressed as

$$\dot{\boldsymbol{\theta}}_t = -\eta \nabla_{\boldsymbol{\theta}} \boldsymbol{f}(\boldsymbol{\mathcal{X}}, \boldsymbol{\theta}_t)^\mathsf{T} \nabla_{\boldsymbol{f}(\boldsymbol{\mathcal{X}}, \boldsymbol{\theta}_t)} \mathcal{L} \tag{1}$$

$$\dot{\boldsymbol{f}}(\boldsymbol{\mathcal{X}}, \boldsymbol{\theta}_t) = \nabla_{\boldsymbol{\theta}} \boldsymbol{f}(\boldsymbol{\mathcal{X}}, \boldsymbol{\theta}_t) \dot{\boldsymbol{\theta}}_t = -\eta \boldsymbol{\Theta}_t(\boldsymbol{\mathcal{X}}, \boldsymbol{\mathcal{X}}) \nabla_{\boldsymbol{f}(\boldsymbol{\mathcal{X}}, \boldsymbol{\theta})} \mathcal{L} \tag{2}$$

where the matrix $\boldsymbol{\Theta}_t(\boldsymbol{\mathcal{X}}, \boldsymbol{\mathcal{X}}) \in \mathbb{R}^{NK \times NK}$ is the **Neural Tangent Kernel** at time step $t$, defined as the covariance structure of the Jacobian of the parameters over all training samples:

$$\boldsymbol{\Theta}_t(\boldsymbol{\mathcal{X}}, \boldsymbol{\mathcal{X}}) = \nabla_{\boldsymbol{\theta}} \boldsymbol{f}(\boldsymbol{\mathcal{X}}, \boldsymbol{\theta}_t) \nabla_{\boldsymbol{\theta}} \boldsymbol{f}(\boldsymbol{\mathcal{X}}, \boldsymbol{\theta}_t)^\mathsf{T}. \tag{3}$$

For infinitely wide networks, the NTK exactly captures the output space dynamics through training, and $\boldsymbol{\Theta}_t(\boldsymbol{\mathcal{X}}, \boldsymbol{\mathcal{X}})$ remains constant throughout. Lee et al. (2019a) have shown that neural networks of

any depth tend to follow the linearized training dynamics as predicted by the NTK. Moreover, we can approximate the outputs of the network linearly through a one-step Taylor expansion given input $\boldsymbol{x}$ as:

$$\boldsymbol{f}^-(\boldsymbol{x},\boldsymbol{\theta}_t) = \boldsymbol{f}(\boldsymbol{x},\boldsymbol{\theta}_0) + \nabla_{\boldsymbol{\theta}}\boldsymbol{f}(\boldsymbol{x},\boldsymbol{\theta}_0)\boldsymbol{\omega}_t \tag{4}$$

where $\boldsymbol{\omega}_t = \boldsymbol{\theta}_t - \boldsymbol{\theta}_0$ is the change in parameters from their initial values. The first term of Equation 4 is constant while the second term captures the dynamics of the initial outputs during training. Substituting $\boldsymbol{f}^-$ for $\boldsymbol{f}$ in Equations 1 and 2, the dynamics of the linearized gradient flow become

$$\dot{\boldsymbol{\omega}}_t = -\eta \nabla_{\boldsymbol{\theta}}\boldsymbol{f}(\boldsymbol{\mathcal{X}},\boldsymbol{\theta}_0)^\intercal \nabla_{\boldsymbol{f}^-(\boldsymbol{\mathcal{X}},\boldsymbol{\theta}_t)}\mathcal{L} \tag{5}$$

$$\dot{\boldsymbol{f}}^-(\boldsymbol{x},\boldsymbol{\theta}_t) = -\eta\boldsymbol{\Theta}(\boldsymbol{x},\boldsymbol{\mathcal{X}})\nabla_{\boldsymbol{f}^-(\boldsymbol{\mathcal{X}},\boldsymbol{\theta}_t)}\mathcal{L} \tag{6}$$

Under MSE loss, the above ODEs have closed-form solutions as

$$\boldsymbol{\omega}_t = -\nabla_{\boldsymbol{\theta}}\boldsymbol{f}(\boldsymbol{\mathcal{X}},\boldsymbol{\theta}_0)^\intercal \boldsymbol{\Theta}_0^{-1}\left(\boldsymbol{I} - e^{-\eta\boldsymbol{\Theta}_0 t}\right)(\boldsymbol{f}(\boldsymbol{\mathcal{X}},\boldsymbol{\theta}_0) - \boldsymbol{\mathcal{Y}}) \tag{7}$$

$$\boldsymbol{f}^-(\boldsymbol{\mathcal{X}},\boldsymbol{\theta}_t) = \left(\boldsymbol{I} - e^{-\eta\boldsymbol{\Theta}_0 t}\right)\boldsymbol{\mathcal{Y}} + e^{-\eta\boldsymbol{\Theta}_0 t}\boldsymbol{f}(\boldsymbol{\mathcal{X}},\boldsymbol{\theta}_0). \tag{8}$$

In other words, through the tangent kernel and initial outputs of the network, we can compute the training convergence of a linearized neural network before running any gradient descent steps. We will show in Section 3 that, through the Path Kernel, we can reliably approximate the linearized convergence rate of the network in the absence of data and without computing the full NTK.

## 3 THE PATH KERNEL

We will now provide a reformulation of the output behavior of networks in terms of activated paths and their values. This reformulation provides access to a unique decomposition of the NTK that separates the data-dependent output dynamics of the network from those dependent on the architecture and initialization. This reformulation of network behavior in terms of active paths is motivated by Meng et al. (2019) wherein the authors show that gradient descent on networks with homogenous activations may be computed completely in path space. The decomposition provided in this section makes explicit the relationships between the pruning-at-initialization based approaches described in Section 4 and allows for estimation of the convergence behavior of networks at initialization and how pruning affects this behavior.

Let $\boldsymbol{\theta} \in \mathbb{R}^m$ denote the network parameters in vector form and let $\boldsymbol{x} \in \mathbb{R}^d$ denote an input. Assume the network has $K$ output nodes. We define a path from input to output as a binary vector $\boldsymbol{p}$ such that $\boldsymbol{p}_j = 1$ when $\boldsymbol{\theta}_j$ is an edge along path $\boldsymbol{p}$, otherwise $\boldsymbol{p}_j = 0$. Denote by $\mathcal{P}$ the enumerable set of all paths and $\mathcal{P}_{s \to k}$ the subset of paths that go from input node $s$ to output node $k$. Let $P = |\mathcal{P}|$ be the number of paths. Given the enumeration in $\mathcal{P}$, we will abuse notation slightly and refer to paths $\boldsymbol{p}$ by their index $p$ in $\mathcal{P}$. The value of a path $\boldsymbol{p}$ can be calculated as a product of parameters with binary exponents $\boldsymbol{v}_p(\boldsymbol{\theta}) = \prod_{j=1}^m \boldsymbol{\theta}_j^{\boldsymbol{p}_j}$ such that $\boldsymbol{v}_p$ is the product along the $\boldsymbol{\theta}$ weights that are in path $\boldsymbol{p}$. The activation status of a path $\boldsymbol{p}$ is $a_p(\boldsymbol{x},\boldsymbol{\theta}) = \prod_{\{j \,|\, \boldsymbol{p}_j=1\}} \mathbb{I}(o_{\boldsymbol{p}_j}(\boldsymbol{x},\boldsymbol{\theta}) > 0)$. Here $o_{\boldsymbol{p}_j}$ is the output of the hidden node which path $\boldsymbol{p}_j$ passes through immediately after parameter $j$. We can then define the output of the network at node $k$ as

$$\boldsymbol{f}^k(\boldsymbol{x},\boldsymbol{\theta}) = \sum_{s=1}^d \sum_{p \in \mathcal{P}_{s \to k}} \boldsymbol{v}_p(\boldsymbol{\theta})a_p(\boldsymbol{x},\boldsymbol{\theta})\boldsymbol{x}_s. \tag{9}$$

The derivative of $a_p(\boldsymbol{x},\boldsymbol{\theta})$ with respect to $\boldsymbol{\theta}$ will goes to zero in expectation. Using the chain rule, we can break the derivative of the output of the network with respect to $\boldsymbol{\theta}$ into two parts, the partial derivative of the output with respect to the path values and the partial derivative of the path values with respect to the parameters:

$$\nabla_{\boldsymbol{\theta}}\boldsymbol{f}(\boldsymbol{x},\boldsymbol{\theta}) = \frac{\partial\boldsymbol{f}(\boldsymbol{x},\boldsymbol{\theta})}{\partial\boldsymbol{\theta}} = \frac{\partial\boldsymbol{f}(\boldsymbol{x},\boldsymbol{\theta})}{\partial\boldsymbol{v}(\boldsymbol{\theta})}\frac{\partial\boldsymbol{v}(\boldsymbol{\theta})}{\partial\boldsymbol{\theta}} = \boldsymbol{J}_{\boldsymbol{v}}^{\boldsymbol{f}}(\boldsymbol{x})\boldsymbol{J}_{\boldsymbol{\theta}}^{\boldsymbol{v}}. \tag{10}$$

Note the inner product structure formed over paths as $\frac{\partial\boldsymbol{f}(\boldsymbol{x},\boldsymbol{\theta})}{\partial\boldsymbol{v}(\boldsymbol{\theta})} = \boldsymbol{J}_{\boldsymbol{v}}^{\boldsymbol{f}}(\boldsymbol{x}) \in \mathbb{R}^{K \times P}$ and $\frac{\partial\boldsymbol{v}(\boldsymbol{\theta})}{\partial\boldsymbol{\theta}} = \boldsymbol{J}_{\boldsymbol{\theta}}^{\boldsymbol{v}} \in \mathbb{R}^{P \times m}$. The change in output with respect to path values now only depends on the activation status of each path leading to the output and the input to that path. Each entry of this matrix has value

$$(\boldsymbol{J}_{\boldsymbol{v}}^{\boldsymbol{f}}(\boldsymbol{x}))_{k,p} = \left(\frac{\partial\boldsymbol{f}(\boldsymbol{x},\boldsymbol{\theta})}{\partial\boldsymbol{v}(\boldsymbol{\theta})}\right)_{k,p} = \sum_{s=1}^d \mathbb{I}(p \in \mathcal{P}_{s \to k})a_p(\boldsymbol{x},\boldsymbol{\theta})\boldsymbol{x}_s.$$

Similarly, the change in path values with respect to parameters is a function of only the parameters and their relational structure through paths:

$$(\boldsymbol{J_\theta^v})_{p,j} = \begin{cases} \frac{\boldsymbol{v}_p}{\boldsymbol{\theta}_j} & \boldsymbol{p}_j = 1 \\ 0 & \text{otherwise.} \end{cases}$$

Given this reparameterization of the output of the network in terms of activated paths and their values, we see the NTK similarly decomposes nicely along this structure:

$$\boldsymbol{\Theta}(\boldsymbol{x}, \boldsymbol{x}) = \nabla_\theta \boldsymbol{f}(\boldsymbol{x}, \boldsymbol{\theta}) \nabla_\theta \boldsymbol{f}(\boldsymbol{x}, \boldsymbol{\theta})^\intercal$$
$$= \boldsymbol{J_v^f}(\boldsymbol{x}) \boldsymbol{J_\theta^v} (\boldsymbol{J_\theta^v})^\intercal \boldsymbol{J_v^f}(\boldsymbol{x})^\intercal$$
$$= \boldsymbol{J_v^f}(\boldsymbol{x}) \boldsymbol{\Pi_\theta} \boldsymbol{J_v^f}(\boldsymbol{x})^\intercal$$

where $\boldsymbol{\Pi_\theta} = \boldsymbol{J_\theta^v} \boldsymbol{J_\theta^{v\intercal}}$ is the **Path Kernel**. The Path Kernel $\boldsymbol{\Pi_\theta}$ is positive semidefinite with maximal values on the diagonal $\boldsymbol{\Pi_\theta}(p, p) = \sum_{j=1}^m (\frac{\boldsymbol{v}_p(\boldsymbol{\theta})}{\boldsymbol{\theta}_j})^2$ and off-diagonal elements $\boldsymbol{\Pi_\theta}(p, p') = \sum_{j=1}^m (\frac{\boldsymbol{v}_p(\boldsymbol{\theta})}{\boldsymbol{\theta}_j})(\frac{\boldsymbol{v}_{p'}(\boldsymbol{\theta})}{\boldsymbol{\theta}_j})$. We can view $\boldsymbol{\Pi_\theta}$ as a covariance matrix on the weighted paths defined by the network architecture and parameter initialization. Note that $\boldsymbol{J_v^f}(\boldsymbol{x})$ entirely captures the dependence of $f$ on the input by choosing which paths are active and re-weighting by the input, while $\boldsymbol{\Pi_\theta}$ is completely determined by the architecture and initialization. Therefore, we can expand our one-sample NTK to the entire training set through the appropriate expansion of dimensions such that $\boldsymbol{J_v^f}(\boldsymbol{\mathcal{X}}) \in \mathbb{R}^{NK \times P}$.

In the following section, we will show that the Path Kernel decomposition of the NTK allows us to approximate, at initialization, the convergence behavior of the network during training. Additionally, we find we can compute the trace of the path kernel efficiently through an implicit computation over the parameter gradients of a particular loss function. This trace computation serves as an approximation to full eigenstructure of the NTK.

### 3.1 NEURAL TANGENT KERNEL CONVERGENCE

We can decompose $\boldsymbol{J_v^f}(\boldsymbol{\mathcal{X}}) = \boldsymbol{VDW^\intercal}$ and $\boldsymbol{\Pi_\theta} = \boldsymbol{USU^\intercal}$ and rewrite the NTK in Equation 3 as

$$\boldsymbol{\Theta}_t(\boldsymbol{\mathcal{X}}, \boldsymbol{\mathcal{X}}) = \boldsymbol{J_v^f}(\boldsymbol{\mathcal{X}}) \boldsymbol{\Pi}_{\theta_t} \boldsymbol{J_v^f}(\boldsymbol{\mathcal{X}})^\intercal \tag{11}$$
$$= \boldsymbol{VDW^\intercal USU^\intercal WDV^\intercal} \tag{12}$$
$$= \boldsymbol{VDU'SU'^\intercal DV^\intercal}. \tag{13}$$

From this perspective, we see the NTK acts as a rotation of inputs onto a set of paths along with an input-weighting of those paths, a computation of similarity in path space, followed by a rotation of the result back to output space. The eigenvectors of the Path Kernel $\boldsymbol{\Pi}_{\theta_t}$ are a formal sum of paths with the eigenvector corresponding to the largest eigenvalue having interpretation as a unit-weighted set of paths producing highest flow through the network. The eigenstructure of the entire NTK is therefore determined by how eigenvectors of $\boldsymbol{\Pi}_{\theta_t}$ are mapped onto by the eigenvectors of $\boldsymbol{J_v^f}(\boldsymbol{\mathcal{X}})$.

**Theorem 1.** *Let $\lambda_i$ be the eigenvalues of $\boldsymbol{\Theta}_t(\boldsymbol{\mathcal{X}}, \boldsymbol{\mathcal{X}})$, $\nu_i$ the eigenvalues of $\boldsymbol{J_v^f}(\boldsymbol{\mathcal{X}})$, and $\pi_i$ the eigenvalues of $\boldsymbol{\Pi}_{\theta_t}$. Then $\lambda_i \leq \nu_i \pi_i$ and $\sum_{i=1}^{NK} \lambda_i \leq \sum_{i=1}^{NK} \nu_i \pi_i$.*

The proof of this theorem is given in the Appendix. Under the NTK assumptions of infinite width or infinite parameter scale (Woodworth et al., 2020), Theorem 2 provides bounds on the network training convergence through the linearized dynamics in Equations 5 and 6. Specifically, if $\boldsymbol{\Theta}_0$ can be diagonalized by eigenfunctions with corresponding eigenvalues $\lambda_i$, then the exponential $e^{-\eta \boldsymbol{\Theta}_0 t}$ has the same eigenfunctions with eigenvalues $e^{-\eta \lambda_i t}$. Therefore, $\sum_{i=1}^{NK} \nu_i \pi_i$ provides an estimate of the network's training convergence at initialization as an upper bound on the scale of $e^{-\eta \boldsymbol{\Theta}_0 t}$ which drives convergence in Equations 7 and 8. In Section 5.1, we show that this estimate does in fact predict convergence on a wide range of finite network architectures, allowing us to predict the convergence performance of a number of initialization pruning algorithms.

The effect of $\boldsymbol{J_v^f}(\boldsymbol{\mathcal{X}})$ on the eigenvalues of the NTK will be a scaling of the eigenvalues of $\boldsymbol{\Pi_\theta}$. If $\nu_1$ is small, the input data map onto numerous paths within the network, increasing complexity and slowing convergence. Conversely, large $\nu_1$ implies fewer paths are activated by the input, leading to

lower complexity and faster convergence. In other words, the interaction between $\boldsymbol{J}_v^f(\boldsymbol{\mathcal{X}})$ and $\boldsymbol{\Pi}_{\boldsymbol{\theta}}$ will be high if they share eigenvectors. See the Section A.2 for further discussion of the effect of $\boldsymbol{J}_v^f(\boldsymbol{\mathcal{X}})$ on the covariance structure of the network.

Given an initialized network, we can approximate this training convergence as the trace of the Path Kernel

$$\mathrm{Tr}(\boldsymbol{\Pi}_{\boldsymbol{\theta}}) = \sum_{p=1}^{P} \boldsymbol{\Pi}_{\boldsymbol{\theta}}(p, p) = \sum_{p=1}^{P} \sum_{j=1}^{m} \left( \frac{\boldsymbol{v}_p(\boldsymbol{\theta})}{\boldsymbol{\theta}_j} \right)^2. \tag{14}$$

This trace can be computed efficiently leveraging an implicit computation over the network's gradients. To see this, define the loss function

$$\mathcal{R}_{\mathrm{PK}}(\boldsymbol{x}, \boldsymbol{\theta}) = \mathbb{1}^{\intercal} \left( \prod_{l=1}^{L} \boldsymbol{\theta}_l^2 \right) \boldsymbol{x}. \tag{15}$$

Letting $\boldsymbol{x} = \mathbb{1}$ the vector of 1's and noting that $\boldsymbol{v}_p(\boldsymbol{\theta}^2) = \prod_{j=1}^{m} \boldsymbol{\theta}_j^{2\boldsymbol{p}_j} = \boldsymbol{v}_p(\boldsymbol{\theta})^2$, we can compute the gradient of this loss

$$\frac{\partial \mathcal{R}_{\mathrm{PK}}(\mathbb{1}, \boldsymbol{\theta})}{\partial \boldsymbol{\theta}_j^2} = \sum_{p=1}^{P} \frac{\boldsymbol{v}_p(\boldsymbol{\theta}^2)}{\boldsymbol{\theta}_j^2}$$

$$\sum_{j=1}^{m} \frac{\partial \mathcal{R}_{\mathrm{PK}}(\mathbb{1}, \boldsymbol{\theta})}{\partial \boldsymbol{\theta}_j^2} = \sum_{p=1}^{P} \sum_{j=1}^{m} \left( \frac{\boldsymbol{v}_p(\boldsymbol{\theta})}{\boldsymbol{\theta}_j} \right)^2 = \mathrm{Tr}(\boldsymbol{\Pi}_{\boldsymbol{\theta}})$$

## 4 PRUNING AND THE PATH KERNEL

The decomposition of the NTK into data-dependent and architecture-dependent pieces provides a generalized view through which to analyze prior approaches to pruning at initialization. Nearly all of the recent techniques for pruning at initialization are derived from a notion of feature **saliency**, a measure of importance of particular parameters with respect to some feature of the network. Letting $F$ represent this feature, we define saliency as the Hadamard product:

$$S(\boldsymbol{\theta}) = \frac{\partial F}{\partial \boldsymbol{\theta}} \odot \boldsymbol{\theta}.$$

The following methods for pruning at initialization may all be viewed as a type of saliency measure of network parameters.

### 4.1 SNIP

Perhaps the most natural feature of the network to target is the loss. This approach, known as skeletonization (Mozer & Smolensky, 1989) and recently reintroduced as SNIP (Lee et al., 2019b), scores network parameters based on their relative contribution to the loss

$$S_{\mathrm{SNIP}}(\boldsymbol{\theta}) = \frac{\partial \mathcal{L}}{\partial \boldsymbol{\theta}} \odot \boldsymbol{\theta}.$$

While intuitive, this pruning approach–which we will refer to as SNIP–is overexposed to scale differences within the network. In other words, large differences in the magnitude of parameters can saturate the gradient, leading to unreliable parameter saliency values, potentially resulting in layer collapse (Tanaka et al., 2020). We can rewrite the SNIP score using our path-based notation:

$$S_{\mathrm{SNIP}}(\boldsymbol{\theta}) = \frac{\partial \mathcal{L}(\boldsymbol{f}(\boldsymbol{\mathcal{X}}, \boldsymbol{\theta}), \boldsymbol{\mathcal{Y}})}{\partial \boldsymbol{\theta}} = \frac{\partial \mathcal{L}(\boldsymbol{f}(\boldsymbol{\mathcal{X}}, \boldsymbol{\theta}), \boldsymbol{\mathcal{Y}})}{\partial \boldsymbol{f}(\boldsymbol{\mathcal{X}}, \boldsymbol{\theta})} \frac{\partial \boldsymbol{f}(\boldsymbol{\mathcal{X}}, \boldsymbol{\theta})}{\partial \boldsymbol{v}(\boldsymbol{\theta})} \frac{\partial \boldsymbol{v}(\boldsymbol{\theta})}{\partial \boldsymbol{\theta}}$$

$$= \frac{\partial \mathcal{L}(\boldsymbol{f}(\boldsymbol{\mathcal{X}}, \boldsymbol{\theta}), \boldsymbol{\mathcal{Y}})}{\partial \boldsymbol{f}(\boldsymbol{\mathcal{X}}, \boldsymbol{\theta})} \boldsymbol{J}_v^f(\boldsymbol{\mathcal{X}}) \boldsymbol{J}_{\boldsymbol{\theta}}^v.$$

Clearly, SNIP scoring is dependent on network paths, as network outputs are fully described by the combination of paths with the activation structure of those paths determined by input $\boldsymbol{x}$. The precise interaction between paths and the loss is dependent on the loss function, but it is clear from the $\boldsymbol{J}_{\boldsymbol{\theta}}^v$

term that the loss will be more sensitive to high-valued paths than low-valued paths. This points to the source of overexposure to scale differences within the network, wherein particular high-valued paths are likely to dominate the loss signal, especially if initialization parameter scales are large (Lee et al., 2020). The effect of large scale or variance in parameter initialization is multiplicative along paths, and given the direct dependence of the network output on paths, the loss is likely to be overexposed to the highest-magnitude paths.

## 4.2 SYNFLOW

Tanaka et al. (2020) proposed **SynFlow**, an algorithmic approach to pruning neural networks without data at initialization. SynFlow also relies on saliency, but computes saliency with respect to a particular loss function:

$$
\mathcal{R}_{\text{SF}}(\mathbb{1}, \boldsymbol{\theta}) = \mathbb{1}^{\intercal} \left( \prod_{l=1}^{L} |\boldsymbol{\theta}_l| \right) \mathbb{1} \tag{16}
$$

such that the synaptic saliency score is $S_{\text{SF}}(\boldsymbol{\theta}) = \frac{\partial \mathcal{R}_{\text{SF}}(\mathbb{1}, \boldsymbol{\theta})}{\partial \boldsymbol{\theta}} \odot \boldsymbol{\theta}$ which the authors call the "Synaptic Flow". Note the structural similarity of Equation 16 to Equation 15; we can view SynFlow as an $\ell_1$ approximation to the trace of the Path Kernel and thus an approximation of the sum of its eigenvalues.

This observation leads to our proposal of two variants of the SynFlow pruning approach. This first variant, which we call **SynFlow-L2**, is derived from the observation that SynFlow is an $\ell_1$ approximation of the effect of the parameters on the Path Kernel structure. We score parameters in this variant as $S_{\text{SF-L2}}(\boldsymbol{\theta}) = \frac{\partial \mathcal{R}_{\text{PK}}(\mathbb{1}, \boldsymbol{\theta})}{\partial \boldsymbol{\theta}^2} \odot \boldsymbol{\theta}$. Due to the parameter squaring, we expect SynFlow-L2 to score more highly parameters along paths with high weight magnitudes relative to SynFlow. Given the squared interaction of path values defining the eigenstructure of the Path Kernel (Equation 14), we expect the Synflow-L2 pruning approach to better capture the dominant training dynamics as the architecture of the network becomes wider and better approximates the NTK assumptions of infinite width. We find empirical evidence for this fact in Section 5.2.

The second SynFlow variant comes from observing the effect of the free input variable $\boldsymbol{x}$ in Equation 15. The SynFlow algorithm sets $\boldsymbol{x} = \mathbb{1}$ which has the effect of computing scores over all possible paths. But not all paths may be used given a particular input distribution (see Section A.2 for an analysis of this fact for a 2-layer network). Therefore, if we have knowledge at the time of initialization of which input dimensions are more task-relevant, we can weight $\boldsymbol{x}$ to reflect this discrepancy in relevance across input dimensions. This re-weighting increases the input-weighted value of paths as calculated by Equations 15 and 16. For the datasets used in Section 5, we assume we do not have *a priori* knowledge of which input dimensions are most relevant for classification success on each dataset. Therefore, we use the mean $\boldsymbol{\mu}$ as a proxy for the "importance" of each input dimension under the assumption that important paths may be associated to high-magnitude input dimensions in expectation. We refer to this distributional variation of SynFlow as **SynFlow-Dist** defined as $S(\boldsymbol{\mu}, \boldsymbol{\theta})_{\text{SF-D}} = \frac{\partial \mathcal{R}_{\text{SF}}(\boldsymbol{\mu}, \boldsymbol{\theta})}{\partial \boldsymbol{\theta}} \odot \boldsymbol{\theta}$ and the distributional variation of SynFlow-L2 as **SynFlow-L2-Dist** defined as $S(\boldsymbol{\mu}, \boldsymbol{\theta})_{\text{SF-L2-D}} = \frac{\partial \mathcal{R}_{\text{PK}}(\boldsymbol{\mu}, \boldsymbol{\theta})}{\partial \boldsymbol{\theta}^2} \odot \boldsymbol{\theta}$.

## 4.3 GRASP

Wang et al. (2020) make use of the NTK by pruning according to a measure of preservation of gradient flow after pruning by analyzing the change in loss $\Delta \mathcal{L}$ after pruning, represented as a small deviation $\boldsymbol{\delta}$ in the weight:

$$
\begin{aligned}
S_{\text{GraSP}}(\boldsymbol{\delta}) &= \Delta \mathcal{L}(\boldsymbol{\theta}_0 + \boldsymbol{\delta}) - \Delta \mathcal{L}(\boldsymbol{\theta}_0) \\
&= 2\boldsymbol{\delta}^{\intercal} \nabla_{\boldsymbol{\theta}}^2 \mathcal{L}(\boldsymbol{\theta}_0) \nabla_{\boldsymbol{\theta}} \mathcal{L}(\boldsymbol{\theta}_0)) + \mathcal{O}(\|\boldsymbol{\delta}\|_2^2) \\
&= 2\boldsymbol{\delta}^{\intercal} \boldsymbol{H} \nabla_{\boldsymbol{\theta}} \mathcal{L}(\boldsymbol{\theta}_0) + \mathcal{O}(\|\boldsymbol{\delta}\|_2^2).
\end{aligned}
$$

where $\boldsymbol{H} = \nabla_{\boldsymbol{\theta}}^2 \mathcal{L}(\boldsymbol{\theta}_0)$ is the Hessian. When $\boldsymbol{H} = \boldsymbol{I}$, this score recovers SNIP, up to absolute value ($|\boldsymbol{\delta}^{\intercal} \nabla \mathcal{L}(\boldsymbol{\theta}_0)|$). This pruning approach, called GraSP, scores each weight according to the change in gradient flow after pruning the weight with the Hessian approximating the dependencies between weights with respect to the gradient. We can also embed GraSP within the saliency framework as $S_{\text{GRaSP}}(\boldsymbol{\theta}) = - \left( \boldsymbol{H} \frac{\partial \mathcal{L}}{\partial \boldsymbol{\theta}} \right) \odot \boldsymbol{\theta}$.

We can rewrite $\boldsymbol{H}$ using our path notation as

$$\boldsymbol{H} = \nabla_{\boldsymbol{\theta}}^2 \mathcal{L} = \mathrm{mat}(\boldsymbol{H}_{\boldsymbol{\theta}}^{\boldsymbol{v}} \boldsymbol{J}_{\boldsymbol{v}}^{\boldsymbol{f}}(\boldsymbol{x})^{\mathsf{T}} \nabla_{\boldsymbol{f}} \mathcal{L}) + \boldsymbol{J}_{\boldsymbol{\theta}}^{\boldsymbol{v}\mathsf{T}} \boldsymbol{J}_{\boldsymbol{v}}^{\boldsymbol{f}}(\boldsymbol{x})^{\mathsf{T}} \nabla_{\boldsymbol{f}} \mathcal{L} \boldsymbol{J}_{\boldsymbol{v}}^{\boldsymbol{f}}(\boldsymbol{x}) \boldsymbol{J}_{\boldsymbol{\theta}}^{\boldsymbol{v}}. \qquad (17)$$

The function $\mathrm{mat}(\boldsymbol{a})$ takes vector $\boldsymbol{a} \in \mathbb{R}^{n^2}$ and forms a matrix $\boldsymbol{A} \in \mathbb{R}^{n \times n}$ and $\boldsymbol{H}_{\boldsymbol{\theta}}^{\boldsymbol{v}} = \left(\frac{\partial}{\partial \boldsymbol{\theta}} \boldsymbol{J}_{\boldsymbol{\theta}}^{\boldsymbol{v}\mathsf{T}}\right) \in \mathbb{R}^{m^2 \times p}$ is the second derivative of the path value matrix. See Section A.3 for the derivation. Equation 17 makes clear the relationship between paths and the GraSP scores for parameters. GraSP scoring takes SNIP scores and re-scales these scores by their interacting path values. The $\boldsymbol{H}_{\boldsymbol{\theta}}^{\boldsymbol{v}}$ is especially interesting as it reflects the second-derivative of the path values according to pairs of parameters along paths. In short, the addition of the Hessian in GraSP scoring makes the pruning scores much more path dependent compared to SNIP, albeit in a convoluted manner depending on both first and second-order effects of parameter changes on path values.

## 5 Experiments and Results

We now bring together a number of ideas from the previous sections to show the empirical validity of the path reformulation of the NTK presented in the preceding sections for analyzing initialization pruning approaches. We begin by showing that the trace of the Path Kernel is indeed a reasonable approximation of the linearized convergence dynamics approximated by Equations 1 and 2 and observe how different pruners affect these dynamics. Given the similarities between SynFlow and the computation of $\mathrm{Tr}(\boldsymbol{\Pi}_{\boldsymbol{\theta}})$, we proposed in Section 4 three SynFlow-like pruning variants for pruning at initialization with and without knowledge of the data distribution. We test each of these variants on a number of model-dataset combinations and observe, as predicted in Section 4, effects of model width on these pruning variants.

### 5.1 Approximating Convergence Dynamics

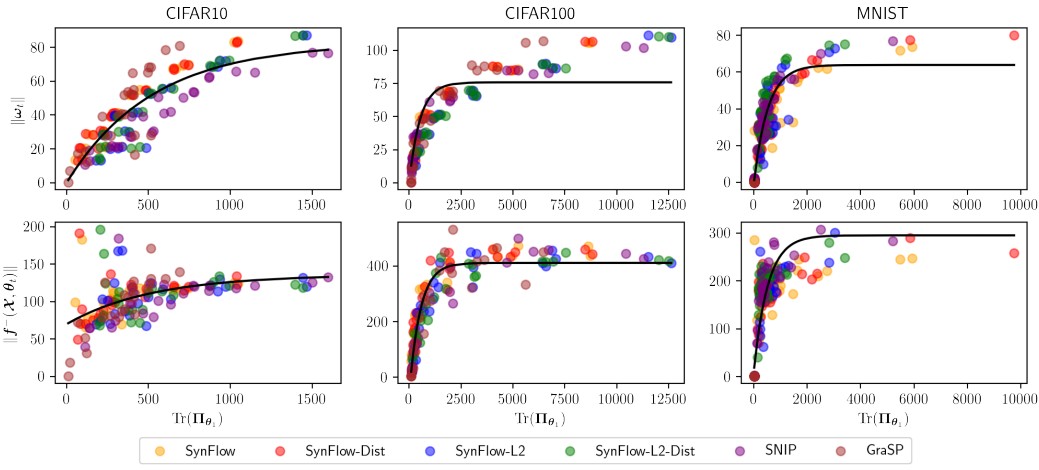

Figure 1: **Aggregate weight change from initialization and output evolution aligns with Path Kernel estimate near initialization**. Each point represents a particular FC-$X$, ResNet-20 or WideResNet-20 model pruned at initialization at a range of compression ratios [0.5-3.0], colored by each pruner. The plotted curve is the continuous-time convergence estimation of Equations 1 and 2.

Section 3.1 introduced the trace of the path kernel at initialization $\mathrm{Tr}(\boldsymbol{\Pi}_{\boldsymbol{\theta}_0})$ as an approximation of the eigenstructure of the NTK at initialization $\boldsymbol{\Theta}_{\boldsymbol{\theta}_0}$. As detailed in Equations 7 and 8, this eigenstructure will dominate the convergence dynamics of the network under the NTK assumptions. Figure 1 plots a number of architectures at varying compression ratios by pruner along with the exponential curve estimated as $\|\boldsymbol{\omega}_t\| = a\left(1 - e^{-\eta\,\mathrm{Tr}(\boldsymbol{\Pi}_{\boldsymbol{\theta}_1})t}\right)$ and $\|\boldsymbol{f}^-(\boldsymbol{\mathcal{X}}, \boldsymbol{\theta}_t)\| = \left(1 - e^{-\eta\,\mathrm{Tr}(\boldsymbol{\Pi}_{\boldsymbol{\theta}_1})t}\right)a + (e^{-\eta\,\mathrm{Tr}(\boldsymbol{\Pi}_{\boldsymbol{\theta}_1})t})b$. Here, $\eta = 0.001$ and free parameters $a$ and $b$ for each point capture the multiplicative constants in Equations 1 and 2 fit at epoch 60. We choose the Path Kernel trace at epoch 1 for better visualization clarity as $\boldsymbol{\Pi}_{\boldsymbol{\theta}_0})$ has much larger variance (see Figure 2). Clearly, the Path Kernel acts as a useful

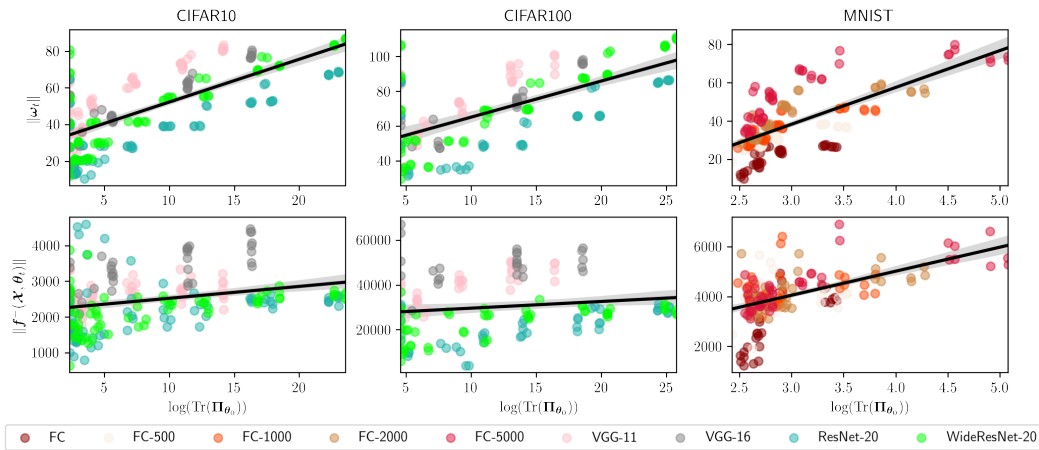

Figure 2: **Aggregate weight change from initialization and output evolution is predicted by** $\text{Tr}(\mathbf{\Pi}_{\boldsymbol{\theta}_0})$. Each point corresponds to a particular model pruned at initialization at a range of compression ratios [0.5-3.0], colored by model. The line of best fit is estimated on $\text{Tr}(\mathbf{\Pi}_{\boldsymbol{\theta}_0})$ versus each aggregate convergence value ($\|\boldsymbol{\omega}_t\|$ and $\|\boldsymbol{f}^-(\boldsymbol{\mathcal{X}}, \boldsymbol{\theta}_t)\|$) with coefficient of determination inset.

proxy for the underlying NTK contribution to the estimation of the linearized dynamics as the empirical models across a range of seeds, compression ratios, and datasets lie along the predicted curve. The theoretically-approximated curve under-estimates the parameter and output changes for high values of $\text{Tr}(\mathbf{\Pi}_{\boldsymbol{\theta}_1})$. This is expected as the multiplicative data-dependent effects on the global NTK eigenstructure are missing from the Path Kernel estimation. Figure 1 also depicts the effects of the pruning approaches on convergence dynamics as described in Section 4. The L2 SynFlow variants align more closely with the predicted curve than the other pruning approaches. SNIP tends to produce models that have relatively larger Path Kernel eigenvalues while GraSP produces the opposite. The SynFlow pruners tend to lie between these extremes across most of the model-dataset combinations.

Figure 2 depicts the log-linear relationship between the Path Kernel eigenstructure and the estimated linearized dynamics across models. Each point again represents a model at varying compression ratios. We fit a simple linear model to the natural logarithm of the trace of the Path Kernel at initialization, predicting each model's associated aggregate convergence value ($\|\boldsymbol{\omega}_t\|$ or $\|\boldsymbol{f}^-(\boldsymbol{\mathcal{X}}, \boldsymbol{\theta}_t)\|$). We observe a generally linear relationship between $\log(\text{Tr}(\mathbf{\Pi}_{\boldsymbol{\theta}_0}))$ with different models exhibiting approximately equivalent slopes with different intercepts. These analyses point towards the general power of the Path Kernel in predicting convergence both within and across models and across compression ratios or pruning approaches.

## 5.2 PATH KERNEL PRUNING VARIANTS

Figure 3 plots the Top-1 Accuracy of a number of models on three datasets pruned at a number of compression ratios, averaged across three random initializations. We leverage the simplicity of MNIST to train four fully-connected networks with six hidden layers that vary in width between models. Model FC has hidden layers of width 100 while the other models FC-$W$ have hidden layers of width $W$. As we can see from the top row of Figure 3, the L2 variants of SynFlow underperform on skinny networks. However, SynFlow-L2 becomes much more competitive as network width increases, surpassing SynFlow at high compression ratios on FC-1000 and FC-2000. The distributional variants generally underperform the non-distributional pruners. Given the relative sparsity of MNIST in its input dimensionality, one might expect a number of paths picked out by SynFlow or SynFlow-L2 to be irrelevant to the dataset. This does not seem to be the case given the underperformance of these distributional pruners. This observation provides further motivation for focusing on the path value term of the Path Kernel for estimating network training dynamics and generalization performance.

The second and third rows of Figure 3 again plot the averaged Top-1 Accuracy across three random initializations on CIFAR10 and CIFAR100, respectively. Notably, VGG-11 and VGG-16 both suffer from layer collapse at high compression ratios on at least one of the datasets and are generally noisy in their pruning performance. The ResNet20 models are much more stable across compression ratios and datasets. Again, we see an increase in the performance of the SynFlow-L2 pruning approach as we increase model width moving from ResNet20 to WideResNet20 as predicted in Section 4.

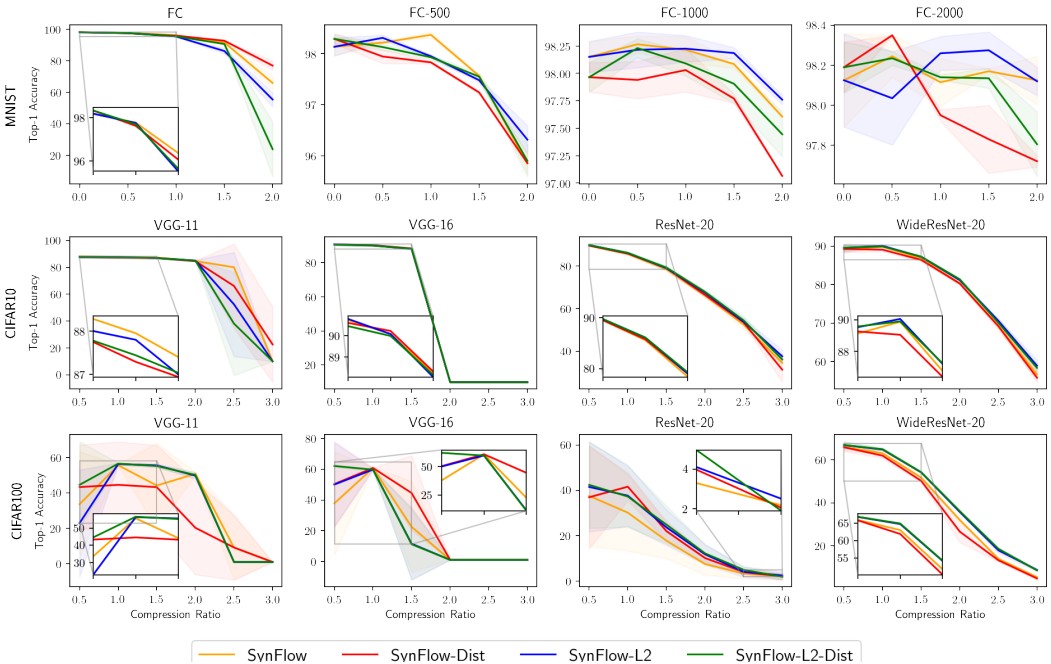

Figure 3: **Top-1 accuracy of models pruned by SynFlow and Path Kernel variants**. Plotted is the mean Top-1 accuracy across three seeds for each combination of variant, model, and dataset. SynFlow-L2 underperforms SynFlow on skinnier networks but outperforms on wider networks as predicted in Section 4. The distributional variants perform better at lower pruning thresholds but generally underperform their non-distributional counterparts.

## 6 DISCUSSION

Recent work on the linear approximation of neural network training dynamics through the Neural Tangent Kernel have shown that these dynamics may be solved for analytically under the assumption of infinite width. However, many practical models currently in use are well-approximated by this theory, despite their finite architectures (Lee et al., 2019a). Leveraging this theory, we showed that for networks with homogeneous activation functions the Neural Tangent Kernel can be decomposed into a product of input-dependent and architecture-dependent terms. Specifically, at initialization, our results show that this decomposition provides an approximation of the network's linearized convergence dynamics without any reference to input data. With this understanding, we applied this theory to the problem of optimal network pruning at initialization and showed that a number of popular initialization pruning approaches may be embedded within this path framework, and that their effects on training dynamics may be understood in terms of their effect on the network's path structure. There are a number of natural routes for future work stemming from this application of the Path Kernel to pruning at initialization. Most pertinent of these is in bounding the data-dependent effects of the NTK eigenstructure as discussed in Theorem 1. An approximation of the data-dependent contribution to the eigenstructure would provide increased accuracy in the approximation of convergence dynamics for any network-dataset pair.

The Path Kernel decomposition likely has many more uses outside of the initialization pruning context presented here. The previous sections have shown the strong bias imported into neural network training and generalization performance by their composite paths. These biases comprise the central features of study in the representation and transfer learning literature. Importantly, we can calculate the Path Kernel for a network at any stage in training, even at convergence. The Path Kernel encodes these representational structures as a type of covariance matrix over composite paths which provides a form that can be further exploited to characterize the implicit representations embedded in networks, compare learned representations across networks, predict the performance of a network given a particular data distribution, or prioritize the presence of particular path-determined biases throughout the course of training.

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

# A APPENDIX

## A.1 PROOF OF THEOREM 1

We repeat the theorem below for convenience:

**Theorem 2.** *Let $\lambda_i$ be the eigenvalues of $\boldsymbol{\Theta}_t(\boldsymbol{\mathcal{X}}, \boldsymbol{\mathcal{X}})$, $\nu_i$ the eigenvalues of $\boldsymbol{J}_{\boldsymbol{v}}^{\boldsymbol{f}}(\boldsymbol{\mathcal{X}})$, and $\pi_i$ the eigenvalues of $\boldsymbol{\Pi}_{\boldsymbol{\theta}_t}$. Then $\lambda_i \leq \nu_i \pi_i$ and $\sum_{i=1}^{NK} \lambda_i \leq \sum_{i=1}^{NK} \nu_i \pi_i$.*

*Proof.* This fact is evident from the decomposition in (12) and (13). $\boldsymbol{\Pi}_{\boldsymbol{\theta}_t}$ is positive semidefinite and will not alter the eigenvalues of $\boldsymbol{J}_{\boldsymbol{v}}^{\boldsymbol{f}}(\boldsymbol{\mathcal{X}})$. To see this, consider eigenvector $\boldsymbol{y}$ of $\boldsymbol{\Theta}_t$. Define $\boldsymbol{y}_{\boldsymbol{V}} = \boldsymbol{V}\boldsymbol{y}$, then $\boldsymbol{y}_{\boldsymbol{V}}^{\mathsf{T}}\boldsymbol{V}\boldsymbol{D}\boldsymbol{W}^{\mathsf{T}}\boldsymbol{\Pi}_{\boldsymbol{\theta}_t}\boldsymbol{W}\boldsymbol{D}\boldsymbol{V}^{\mathsf{T}}\boldsymbol{y}_{\boldsymbol{V}} = \boldsymbol{y}^{\mathsf{T}}\boldsymbol{D}\boldsymbol{U}'\boldsymbol{S}\boldsymbol{U}'^{\mathsf{T}}\boldsymbol{D}\boldsymbol{y}$. $\square$

## A.2 RELATING INPUTS TO PATH ACTIVATION

Arora et al. (2019) showed that under the constant NTK assumption and given labels $\boldsymbol{\mathcal{Y}} \in \mathbb{R}^{NK}$, the training dynamics for squared error loss for a two-layer network may be written:

$$\|\boldsymbol{\mathcal{Y}} - \boldsymbol{f}(\boldsymbol{\mathcal{X}}, \boldsymbol{\theta}_t)\|_2 = \sqrt{\sum_{i=1}^{N}(1 - \eta\lambda_i)^{2t}(\boldsymbol{v}_i^{\mathsf{T}}\boldsymbol{\mathcal{Y}})^2} \pm \epsilon \tag{18}$$

where $\epsilon$ is a bounded error term and $\boldsymbol{H}^{\infty} = \boldsymbol{V}\boldsymbol{\Lambda}\boldsymbol{V}^{\mathsf{T}} = \sum_{i=1}^{N}\lambda_i\boldsymbol{v}_i\boldsymbol{v}_i^{\mathsf{T}}$ is the eigendecomposition of the **Gram matrix** defined as

$$\boldsymbol{H}_{i,j}^{\infty} = \mathbb{E}_{\boldsymbol{w} \sim \mathcal{N}(\boldsymbol{0}, \boldsymbol{I})}[\boldsymbol{x}_i^{\mathsf{T}}\boldsymbol{x}_j \mathbb{I}(\boldsymbol{w}^{\mathsf{T}}\boldsymbol{x}_i \geq 0, \boldsymbol{w}^{\mathsf{T}}\boldsymbol{x}_j \geq 0)] \tag{19}$$

$$= \frac{\boldsymbol{x}_i^{\mathsf{T}}\boldsymbol{x}_j(\pi - \arccos(\boldsymbol{x}_i^{\mathsf{T}}\boldsymbol{x}_j))}{2\pi} \tag{20}$$

for inputs $\boldsymbol{x}_i, \boldsymbol{x}_j$ and column $\boldsymbol{w}$ of $\boldsymbol{\theta}$. Equation 18 shows that the training dynamics of a two-layer network may be approximated as the eigenvalue-weighted sum of the inner product of NTK eigenvectors on the output labels, with convergence being driven by the directions of maximal overlap of network inputs.

In this case of a two-layer, fully-connected network with a single output dimension (19), the effect of $\boldsymbol{J}_{\boldsymbol{v}}^{\boldsymbol{f}}(\boldsymbol{\mathcal{X}})$ on the network output covariance structure becomes clear. Recall the definition of path activation

$$a_p(\boldsymbol{x}, \boldsymbol{\theta}) = \prod_{\{j \mid \boldsymbol{p}_j = 1\}} \mathbb{I}(o_{\boldsymbol{p}_j}(\boldsymbol{x}, \boldsymbol{\theta}) > 0).$$

For a network with ReLU activations, normalized inputs $\boldsymbol{x} \sim \mathcal{N}(\boldsymbol{0}, \boldsymbol{I})$, and weights in the first layer drawn from a unit normal distribution $\boldsymbol{\theta}_1 \sim \mathcal{N}(\boldsymbol{0}, \boldsymbol{I})$, the node-wise output activation function for hidden node $h$ is completely described by the $h$th column of $(\boldsymbol{\theta}_l)_{:,h} = \boldsymbol{w}_h$, and has expectation $\mathbb{E}\left[o_{\boldsymbol{p}_j}(\boldsymbol{x}, \boldsymbol{w}_h)\right] = \mathbb{E}\left[\boldsymbol{w}_h^{\mathsf{T}}\boldsymbol{x}\right] = 0$. Therefore, half of the paths in the path kernel will be activated in expectation since each $\boldsymbol{w}$ is also normally distributed with zero mean. The structure of $\boldsymbol{H}_{i,j}^{\infty}$ is equivalent to the NTK under the architectural assumptions described above. Therefore, the network outputs are completely determined after the inner product on the first layer parameters, and due to the homogenous activations, the value of these outputs is only dependent on the sign of this inner product. The path values determine the output values after the active paths are picked out by the inner products.

## A.3 PATH KERNEL DERIVATION OF HESSIAN

We can re-write a network's Hessian defined as the second derivative of the loss with respect to parameters $\boldsymbol{H} = \nabla_{\boldsymbol{\theta}}^2 \mathcal{L}_{i,j} = \frac{\partial^2 \mathcal{L}}{\partial \boldsymbol{\theta}_i \partial \boldsymbol{\theta}_j}$ in terms of its path decomposition values. Recall the

decomposition of the loss given our path view:

$$\nabla_{\boldsymbol{\theta}}\mathcal{L} = \frac{\partial\mathcal{L}(\boldsymbol{f}(\boldsymbol{x},\boldsymbol{\theta}),\boldsymbol{y})}{\partial\boldsymbol{\theta}} = \frac{\partial\mathcal{L}(\boldsymbol{f}(\boldsymbol{x},\boldsymbol{\theta}),\boldsymbol{y})}{\partial\boldsymbol{f}(\boldsymbol{x},\boldsymbol{\theta})}\frac{\partial\boldsymbol{f}(\boldsymbol{x},\boldsymbol{\theta})}{\partial\boldsymbol{v}(\boldsymbol{\theta})}\frac{\partial\boldsymbol{v}(\boldsymbol{\theta})}{\partial\boldsymbol{\theta}}$$

$$= \frac{\partial\mathcal{L}(\boldsymbol{f}(\boldsymbol{x},\boldsymbol{\theta}),\boldsymbol{y})}{\partial\boldsymbol{f}(\boldsymbol{x},\boldsymbol{\theta})}\boldsymbol{J}_{\boldsymbol{v}}^{\boldsymbol{f}}(\boldsymbol{x})\boldsymbol{J}_{\boldsymbol{\theta}}^{\boldsymbol{v}}$$

$$= \nabla_{\boldsymbol{f}}\mathcal{L}^{\intercal}\boldsymbol{J}_{\boldsymbol{v}}^{\boldsymbol{f}}(\boldsymbol{x})\boldsymbol{J}_{\boldsymbol{\theta}}^{\boldsymbol{v}}.$$

Therefore, we are looking to take the second-derivative of this composition of three functions $\nabla_{\boldsymbol{\theta}}^2\mathcal{L} = \frac{\partial}{\partial\boldsymbol{\theta}}\left(\boldsymbol{J}_{\boldsymbol{\theta}}^{\boldsymbol{v}\intercal}\boldsymbol{J}_{\boldsymbol{v}}^{\boldsymbol{f}}(\boldsymbol{x})^{\intercal}\nabla_{\boldsymbol{f}}\mathcal{L}\right)$ which, using the product rule, becomes

$$\nabla_{\boldsymbol{\theta}}^2\mathcal{L} = \left(\frac{\partial}{\partial\boldsymbol{\theta}}\boldsymbol{J}_{\boldsymbol{\theta}}^{\boldsymbol{v}\intercal}\right)\boldsymbol{J}_{\boldsymbol{v}}^{\boldsymbol{f}}(\boldsymbol{x})^{\intercal}\nabla_{\boldsymbol{f}}\mathcal{L} + \boldsymbol{J}_{\boldsymbol{\theta}}^{\boldsymbol{v}\intercal}\left(\frac{\partial}{\partial\boldsymbol{\theta}}\boldsymbol{J}_{\boldsymbol{v}}^{\boldsymbol{f}}(\boldsymbol{x})^{\intercal}\right)\nabla_{\boldsymbol{f}}\mathcal{L} + \boldsymbol{J}_{\boldsymbol{\theta}}^{\boldsymbol{v}\intercal}\boldsymbol{J}_{\boldsymbol{v}}^{\boldsymbol{f}}(\boldsymbol{x})^{\intercal}\left(\frac{\partial}{\partial\boldsymbol{\theta}}\nabla_{\boldsymbol{f}}\mathcal{L}\right).$$

Define the function $\mathrm{mat}(\boldsymbol{a})$ which takes a vector $\boldsymbol{a} \in \mathbb{R}^{n^2}$ and reshapes it into a matrix in $\mathbb{R}^{n \times n}$. Note that $\left(\frac{\partial}{\partial\boldsymbol{\theta}}\boldsymbol{J}_{\boldsymbol{v}}^{\boldsymbol{f}}(\boldsymbol{x})^{\intercal}\right)$ will go to zero in expectation as each $\frac{\partial}{\partial\boldsymbol{\theta}}a_p(\boldsymbol{x},\boldsymbol{\theta})$ has measure zero everywhere but at a single point. We can now define the Hessian as

$$\nabla_{\boldsymbol{\theta}}^2\mathcal{L} = \mathrm{mat}(\boldsymbol{H}_{\boldsymbol{\theta}}^{\boldsymbol{v}}\boldsymbol{J}_{\boldsymbol{v}}^{\boldsymbol{f}}(\boldsymbol{x})^{\intercal}\nabla_{\boldsymbol{f}}\mathcal{L}) + \boldsymbol{J}_{\boldsymbol{\theta}}^{\boldsymbol{v}\intercal}\boldsymbol{J}_{\boldsymbol{v}}^{\boldsymbol{f}}(\boldsymbol{x})^{\intercal}\left(\frac{\partial}{\partial\boldsymbol{\theta}}\nabla_{\boldsymbol{f}}\mathcal{L}\right)$$

$$= \mathrm{mat}(\boldsymbol{H}_{\boldsymbol{\theta}}^{\boldsymbol{v}}\boldsymbol{J}_{\boldsymbol{v}}^{\boldsymbol{f}}(\boldsymbol{x})^{\intercal}\nabla_{\boldsymbol{f}}\mathcal{L}) + \boldsymbol{J}_{\boldsymbol{\theta}}^{\boldsymbol{v}\intercal}\boldsymbol{J}_{\boldsymbol{v}}^{\boldsymbol{f}}(\boldsymbol{x})^{\intercal}\nabla_{\boldsymbol{f}}\mathcal{L}\boldsymbol{J}_{\boldsymbol{v}}^{\boldsymbol{f}}(\boldsymbol{x})\boldsymbol{J}_{\boldsymbol{\theta}}^{\boldsymbol{v}}.$$

where we have defined $\boldsymbol{H}_{\boldsymbol{\theta}}^{\boldsymbol{v}} = \left(\frac{\partial}{\partial\boldsymbol{\theta}}\boldsymbol{J}_{\boldsymbol{\theta}}^{\boldsymbol{v}\intercal}\right) \in \mathbb{R}^{m^2 \times p}$.

## A.4 EXPERIMENTAL DETAILS

### A.4.1 PRUNING ALGORITHMS

Apart from pruning algorithms already mentioned before (Random, SNIP, GraSP, SynFlow) we also implement the pruning algorithms we defined earlier: **SynFlow-L2**, **SynFlow-L2-Dist** and **SynFlow-Dist** by extending the two step method described in (Tanaka et al., 2020) - scoring and masking parameters globally across the network and pruning those with the lowest scores.

### A.4.2 MODEL ARCHITECTURES

We used standard implementations of VGG-11 and VGG-16 from OpenLTH, and FC-100/500/1000/2000, ResNet-18 and WideResNet18 from PyTorch models using the publicly available, open sourced code from (Tanaka et al., 2020) . As described in (Tanaka et al., 2020) we pruned the convolutional and linear layers of these models as prunable parameters, but not biases nor the parameters involved in batchnorm layers. For convolutional and linear layers, the weights were initialized with a Kaiming normal strategy and biases to be zero.

### A.4.3 TRAINING HYPERPARAMETERS

We provide the hyperparameters we used to train our models as listed in Table 1 and 2.

Table 1: **Hyperparameters for ResNet-20, Wide-ResNet-20, VGG11 and VGG16 models**. These hyperparameters were consistent across pruned and unpruned models and were chosen to optimize performance of the unpruned models. For the pruned models, we pruned for 100 epochs except for SNIP and GraSP, where we only pruned for a single epoch.

|  | ResNet-20 | Wide-ResNet-20 | VGG-11/VGG-16 | |
|  | CIFAR-10/100 | CIFAR-10/100 | CIFAR-10 | CIFAR-100 |
|---|---|---|---|---|
| **Optimizer** | adam | adam | adam | adam |
| **Training Epochs** | 100 | 100 | 160 | 160 |
| **Batch Size** | 64 | 64 | 64 | 64 |
| **Learning Rate** | 0.001 | 0.001 | 0.001 | 0.0005 |
| **Learning Rate Drops** | 60 | 60 | 60,120 | 60,120 |
| **Drop Factor** | 0.1 | 0.1 | 0.1 | 0.1 |
| **Weight Decay** | $10^{-4}$ | $10^{-4}$ | $10^{-4}$ | $10^{-4}$ |

Table 2: **Hyperparameters for the FC, FC-500, FC-1000, FC-2000 models**. Each of the FC models have 6 fully connected layers with the indicated number of parameters in each layer. These hyperparameters were consistent across pruned and unpruned models and were chosen to optimize performance of the unpruned models. For the pruned models, we pruned for 100 epochs except for SNIP and GraSP, where we only pruned for a single epoch.

|  | FC / FC-500 / FC-1000 / FC-2000 | |
|  | MNIST | CIFAR-10/100 |
|---|---|---|
| **Optimizer** | adam | adam |
| **Training Epochs** | 50 | 100 |
| **Batch Size** | 64 | 64 |
| **Learning Rate** | 0.01 | 0.001 |
| **Learning Rate Drops** | - | 60 |
| **Drop Factor** | 0.1 | 0.1 |
| **Weight Decay** | $10^{-4}$ | $10^{-4}$ |

