# OpenReview forum: "A Unified Paths Perspective for Pruning at Initialization"
_ICLR.cc/2021/Conference — Reject_

### Official Review · AnonReviewer1 · 2020-10-26
**Neat NTK perspective, but for what?**

**Rating:** 4
**Confidence:** 3

**Review:**

__summary.__
The paper studies the problem of neural network pruning at initialization through the lens of neural tangent kernels (NTK). As a result, the paper delivers a unified perspective on SNIP, GRASP, and SynFlow. Based on the framework, the paper provides a method to approximate the convergence dynamics of pruned models. The paper also motivates a SynFlow-variant from the theoretical framework.

__overall opinion.__
I could not find a very big empirical benefit from adopting the proposed theoretical perspective. Given the limited novelty and theoretical significance, I do not think this paper makes enough contribution to be shared at ICLR.

__originalty.__
As a matter of fact, I did not find the proposed approach strikingly novel. The fact that the considered pruning methods (which are being unified under the given framework) are all based on the first-order approximations is not very surprising; I would rather say that the fact has been quite well known. The use of first-order approximation itself has been prevalent in the network pruning society, at least since the [optimal brain damage paper](https://papers.nips.cc/paper/250-optimal-brain-damage.pdf) at 1989 and has been used until quite recently, in [layerwise obs](https://arxiv.org/abs/1705.07565) or [lookahead pruning](https://openreview.net/forum?id=ryl3ygHYDB). Also, the combination with the NTK theory for the case of pruning at initialization was already outlined with details in the GRASP paper.

__clarity.__
The paper is clearly written.

__significance.__
I do not see a big merit, especially about the pruning variants. The performance gain of using the proposed variant is nonexistent or marginal, compared to the original SynFlow. The materials presented in section 5.1 seems to be potentially useful, but the paper does not elaborate for what purpose such predictability can be used.

---

> ### Author Response · Authors · 2020-11-20
> **Response to Reviewer 1**
>
> We thank the reviewer for taking the time to read and respond to our paper.
>
> We are a bit disappointed by the lack of specific, material concerns provided by this review that are relevant to the work at hand. The body of this review is composed of high-level opinions which may reflect a lack of engagement with the paper and related literature. These opinions are difficult to adequately address from within the confines of the paper’s scope, but we will attempt to do so below.
>
> Regarding “originality”, this comment deviates substantially from the main themes presented in the paper such that it has little relevance to the work and seems to be primarily a meta-comment on the concept of pruning at initialization rather than a comment on any particular idea of the paper at hand. We acknowledge in the introduction that pruning at initialization has been in existence for nearly as long as neural networks themselves and even cite a review from this era (1993). The novelty of our paper is not in presenting a first-order approximation to pruning, and the fact that many initialization pruning methods are based on first-order approximations has little relevance to the quality of the paper. The additional comment regarding GraSP already using NTK theory for pruning at initialization further suggests a lack of engagement with the primary material presented within the paper, as the novelty claimed in our paper is not in “the combination with the NTK theory for the case of pruning at initialization”. As well, we discuss the GraSP approach at length, providing a reformulation of the approach from our proposed Path Kernel framework.
>
> As discussed in the general rebuttal, we have used the extra allotted page in revision to better frame the benefits of our proposed theoretical framework. We hope that this will help to make the benefits of our paper clearer and will better invite researchers to seriously engage with the paper itself.

---

### Official Review · AnonReviewer3 · 2020-10-28
**Path-NTK (Neural Tangent Kernel) for Pruning at Initialization**

**Rating:** 4
**Confidence:** 5

**Review:**

### Summary
This paper is twofold: 1) the authors propose a way of computing a NTK by decomposing a piecewise linear network into paths; 2) the authors propose to use this decomposition to detect the least useful weights at initialization.

### Details
For convenience, I will call the theoretical part "Path-NTK" in the following.

The main usable result of Path-NTK is a proxy for computing the speed of convergence of the neural network. This result is based on two approximations: 1) the influence of the eigenvalues $\nu_i$, related to the data, is neglected; 2) the sum of the eigenvalues $\lambda_i$ is assumed to reflect the value of each $\lambda_i$ (i.e., if $\sum_i \lambda_i$ is large, then each $\lambda_i$ is large).
The justification of 1 has been made in Annex A.2, but, once again, at the cost of more approximations, that is, inputs are assumed to be N(0, 1), which is not true in MNIST.
The justification of 2 is experimental, and has been made in Section 5.1. However, the graphs are only qualitative, and not quantitative. We can see that the larger $\sum_i \lambda_i$, the faster the convergence is. But there is no quantitative estimation of the convergence rate, which could have been compared to $\sum_i \lambda_i$ in order to justify completely the approximation. And, last but not least, the dataset MNIST is missing in these experiments.

About the pruning experiments, I can see in the graphs at the bottom-right of Figure 2 that the proposed method outperforms slightly the precedent ones, but this seems very subtle to me.

### Significance
Path-NTK is original, and the idea is interesting and probably reusable. However, this is limited to piecewise linear networks, and the only practical result is based on many approximations that have been certainly discussed, but they could be improved.

The Lottery Ticket Hypothesis is a certainly a hot topic, but the specific topic "pruning at initialization" seems to be marginal. Moreover, the overall significance of the results is not encouraging.

### Additional comments
The justification of all approximations made in the theoretical part seems to be itself a research topic.

Edit:
### Rebuttal
Remark 1:
 * Figures are much clearer;
 * about Figure 1, I see that MNIST has been added, which is good;

Remark 2:
 * "We can always normalize inputs such that they are distributed as N(0,1)": this is not true. At best, we can put the mean to 0 and the variance to 1. It does not mean anything in terms of distribution *shape*;
 * I maintain that experiments with MNIST were missing in Figure 1 in the first version of the paper. This is more important than pretended in the rebuttal, because, in the MNIST dataset, the pixels are not N(0, 1) (even when centered and normalized), so MNIST does no meet the approximation made in Annex A.2. However, the results are not too different from those obtained with the other datasets.

The remark about the limitation to piecewise linear NNs has not been correctly addressed by the authors. The setup used in this paper corresponds to "Piecewise linear NNs", which is *really* limiting, since the biases are supposed to be zero. This is very different from "Piecewise *affine* NNs", which are actually widely used (biases can be non-zero). Adding the influence of the biases would break the entire computation made in this paper (or, at least, would necessitate more effort to take it into account). I assert that this limitation is not discussed in the paper, and my attempt to discuss it here has been eluded in the rebuttal.

Despite the new and more fashionable figures the authors presented in the rebuttal, I am disappointed by its lack of accuracy and vagueness, especially when discussing the "piecewise linear NNs". So, I lower my rating.

---

> ### Author Response · Authors · 2020-11-20
> **Response to Reviewer 3**
>
> We thank the reviewer for taking the time to read our paper and for offering a number of insightful comments that should go a long way towards improving the quality of our work.
>
> 1. We acknowledge that the data-dependent eigenstructure of the NTK is ignored in our empirical analyses. One of the benefits of this Path Kernel approach is in its ability to approximate convergence dynamics in a data-independent manner. Empirically, we wished to see how far this approximation could take us. Additionally, we do not see any easy way to estimate these data-dependent terms. Given the results presented in the first draft as well as the additional analyses provided in revision (thanks to this reviewer’s comments, see updated Figures 1 and 2), it is clear that approximating the sum of the NTK eigenvalues from the trace of the Path Kernel is a reasonable estimation. We would have certainly liked to compute the empirical NTK eigenvalues and then compared the Path Kernel estimation to this “ground truth”. However, computing the empirical NTK is a resource-intensive task that becomes impractical as the network size or the dataset size grows. For a moderately small network with about 2-2.5 million parameters and a moderately sized dataset of about 30k data points each with a dimension of 3 X 512 X 512,  computing the empirical kernel can take a day or longer depending on the memory of a single GPU available as well as the processing time taken because of the exploding dimensionality of the output Jacobians and the subsequent multiplication to calculate $J  J^T$ [1]. The fact that the structure of the NTK is so difficult to compute empirically further validates the benefits of our Path Kernel approximation which can be done in a single forward pass through the network. We view the problem of relating the eigenstructure introduced by the data and its interaction with the Path Kernel as a promising direction of future work which could have major implications to the areas of representational and transfer learning. (computing the NTK).
>
> 2. We can always normalize inputs such that they are distributed as N(0,1). In fact, this is considered common or even “good” practice for many datasets, including MNIST. We agree that our graphs of the convergence properties were merely “qualitative”. We have amended this in the revised version of the paper. See Figure 2 in which we plot, across a number of models-dataset pairs, the relationship between the trace of the Path Kernel and the movement of weights and change in output for a number of model instantiations. We also report the correlation coefficients for this relationship. The reviewer claims that MNIST is “missing” from these experiments. However, we did include MNIST in these experiments for the fully-connected models both in our original submission and in the revised version. It is our understanding that the general consensus  within the ML community is that MNIST is considered to be too simplistic for many of the larger models (VGG16, ResNet20, Wide-ResNet20, etc.). However, we have included these fully-connected models trained on MNIST in the correlation analysis described above. Additionally, we have run extra random instantiations to make the differentiation between Synflow and its variants clearer through narrowed error bars. We have also added inset plots for convenience.
>
> Finally, this reviewer remarks: “this is limited to piecewise linear networks”. Indeed, our derivations in the paper are contingent on ReLU activations. We would like to note that ReLU is by far the most common activation function in use today. It is a happy coincidence that the properties of this activation function simplify our Path Kernel derivations. However, this obviously leaves something to be desired in terms of generalization to other activation functions. We note that this same theoretical setup can be used for any activation function. However, more care must be taken when deriving the specific form of the composite Jacobian matrices, and the introduction of another similar partial derivative matrix for the effects of this activation function may be in order. Again, this is an exciting direction for future work and we are delighted this reviewer has noticed as well.
>
> [1] https://github.com/google/neural-tangents/issues/30

---

### Official Review · AnonReviewer4 · 2020-10-29
**In this paper, the authors propose a new kernel named Path Kernel to understand deep neural network training.**

**Rating:** 6
**Confidence:** 3

**Review:**

In this paper, the authors propose a new kernel named Path Kernel to understand deep neural network training. The key idea is to reparameterize the network with respect to the active path in the network. In this way, they can decompose the Tangent Kernel into data-dependent and architecture-dependent pieces. They authors rewrite the formulations of the existing pruning at initialization methods with respect to their Path Kernel and provide some new understandings.

The positive aspects:

1. The proposed Path Kernel is novel.

2. Based on Path Kernel, the authors provide some new understandings to the existing pruning methods.

3. This paper is well written and easy to read.

My concerns are:

1. We know that NTK focus on the small neighbourhood near the initialization. However, in deep neural network training, the weights could be quite far from the initialization. So I am not sure whether it is reasonable to view the existing pruning at initialization methods from the perspective of NTK.

2. I think this paper may be a bit over claimed. The authors provide some new understandings to the existing pruning methods instead of unify them into a single framework.  The authors essentially rewrite the formulations in existing pruning methods with respect to active paths and show that they all have Path Kernel. I think this is not surprising since the network is reparameterized with respect  to the active paths. Such rewriting can be performed on lots of other models besides pruning and we can always show there exists Path Kernel in them.

---

> ### Author Response · Authors · 2020-11-20
> **Response to Reviewer 4**
>
> We thank the reviewer for dedicating the time to read and review this work.
>
> 1. The NTK theory indeed assumes that the NTK itself will remain approximately equivalent to its initialized value throughout the course of training. This is precisely the theoretical benefit provided by the NTK theory. Presently, there are two hypothesized regimes in which the NTK of a network will exhibit such behavior: as the network width becomes large, and as the scale of the network becomes large [1]. Recent work has shown that many practical networks indeed exhibit NTK dynamics that are quite similar to their theoretically predicted behavior [2]. All of this is to say that we are confident that these approximations will be good for a number of our network architectures. We also address this problem directly in the experimental section, showing that the Synflow-L2 pruning method performs relatively worse as the width of the network decreases, as predicted under the NTK theory.
>
> 2. The main purpose of rewriting these pruning methods was to show that contained within each of them is an implicit path-based prior which can be decomposed with our Path Kernel theory. We are not sure that this should be interpreted as a point of criticism, as part of the novelty of this work is in showing this reparameterization exists in a general setting. The Path Kernel perspective provides better focus on the network as a computational graph composed as a set of paths, across which data flows from the input to output layers. This reparameterization is precisely what is useful about our proposed perspective: it generalizes across models and is a useful reparameterization for a number of applications. In our revision, we have included slightly more discussion about how these reparameterizations are helpful in understanding the effects these pruning approaches have on the network's training dynamics.
>
> [1]: Woodworth, Blake, et al. "Kernel and rich regimes in overparametrized models." arXiv preprint arXiv:2002.09277 (2020).
> [2]: Lee, Jaehoon, et al. "Wide neural networks of any depth evolve as linear models under gradient descent." Advances in neural information processing systems. 2019.

---

### Official Review · AnonReviewer2 · 2020-10-30
**A novel perspective on explaining the pruning at initialization**

**Rating:** 6
**Confidence:** 3

**Review:**

This paper proposes the path kernel which decomposes NTK in terms of path kernel. It decouples the NTK into two parts, the data-dependent part, and the data-independent part, which is the Path Kernel.

The new perspective to explain several recent pruning algorithms by Path Kernel is novel and valuable. The experiment of the proposed variation of  SynFlow also coincides with the theory.

Pros: a good theoretical foundation on the proposed algorithm which separates the data-independent part to Path Kernel.

Cons:
1. The experiment's setup is not clear to me,  can I ask what is the formal definition of compression ratio. I thought it might be the original size/ compressed size, however, it can be small than 1 or even 0 in Figure 2 which confused me.

2. Recent researches show that pruning at initialization is not better than random pruning when keeping the non-zero ratio among layers. Could the authors provide a comparison between the proposed algorithm and keep-ratio pruning? Here's the reference.
https://arxiv.org/abs/2009.11094
https://arxiv.org/pdf/2009.08576.pdf

---

> ### Author Response · Authors · 2020-11-20
> **Response to Reviewer 2**
>
> We thank the reviewer for taking the time to review our submission and for their questions.
>
> 1. The compression ratio is defined as: $\log_{10} (\text{number weights in unpruned model}) / (\text{number of weights in pruned model})$. Which is why as we prune models more, we get values in the range of 1 or higher. For an unpruned model, compression_ratio is 0.
>
> 2. Both papers referenced were released on the arXiv within 2 weeks of the ICLR submission deadline, so we were not aware of their contributions while preparing this submission. However, we provide comparisons here. In [1] the authors show that data-independent unstructured pruning at initialization results in random tickets performing at par or better than initial tickets obtained using the lottery ticket framework, SNIP, and GraSP. This observation is further supported in [2] which raises questions about sophisticated pruning techniques and the goal of pruning at initialization itself. Both papers exist due to the existence of a theory gap in our understanding of initialization pruning. Our work can be seen as an initial, consolidated theoretical perspective on pruning which looks to help close this gap, complementing the empirical performance measures in [1], [2]. A full analysis through the lens of the Path Kernel as to why structured random pruning is so effective with respect to the myriad of other pruning methods in existence would deserve its own paper. However, we feel our decomposition points to the source of the surprising outperformance of random methods found in [1] and [2]. The Path Kernel parameterizes the covariance structure among paths in the network. By keeping magnitudes fixed within layers or re-sampling weight magnitudes within layers as is done in [1] and [2], the global covariance structure of the paths should change very little in expectation. If we are to believe the NTK assumptions hold for a given network then, as we have shown in this work, this global paths covariance structure is dominant in determining the properties of the network during training. We are excited to more rigorously flesh out this idea in future work, so we thank the reviewer for bringing these results to our attention.
>
> [1] Su, J., Chen, Y., Cai, T., Wu, T., Gao, R., Wang, L., & Lee, J. D. (2020). Sanity-Checking Pruning Methods: Random Tickets can Win the Jackpot. Advances in Neural Information Processing Systems, 33.
> [2] Frankle, J., Dziugaite, G. K., Roy, D. M., & Carbin, M. (2020). Pruning Neural Networks at Initialization: Why are We Missing the Mark?. arXiv preprint arXiv:2009.08576.

---

### Author Response · Authors · 2020-11-20
**General Rebuttal to All Reviewers**

We would like to thank the four reviewers for taking the time to read and provide feedback on this work. Your comments will go a long way towards improving the quality of this work.

There were two concerns that, in one form or another, were shared across subsets of the reviews: a narrow application area and the marginal results of the pruning methods proposed within the paper. We will provide a general rebuttal to these primary concerns below. We will then respond to each reviewers’ individual comments in their respective threads.

One of the concerns shared primarily by Reviewers 1 and 4 was that the application area of this Path Kernel perspective--pruning at initialization--was too narrow. We believe that a minor change in framing of the major contributions of the paper would alleviate these concerns. In fact, all reviewers noted the novelty and potential benefit of our theoretical approach, but some were overly focused on the merits of our chosen application domain. With the additional allotted page, we were able to expand upon the introduction and discussion sections of the paper, providing much-needed framing of the underlying novelty and versatility of the Path Kernel derivation in future applications. To summarize, the goal of this paper was to introduce the Path Kernel as a novel derivation of the NTK and to show how it may be useful as a theoretical framework for understanding the training dynamics, representational structure, and the interaction between data and architecture within neural networks. To demonstrate that this Path Kernel theory has applicable benefits to deep learning practitioners, we chose pruning at initialization as a first application of this theory. We then showed that the Path Kernel eigenstructure offers a data-independent approximation of the NTK eigenstructure, allowing us to predict the convergence dynamics of networks at initialization. We chose to focus on the problem of pruning at initialization due to a number of recent approaches, like Synflow, that look to prune networks in the absence of data. As well, the primary application domain of the NTK theory has been in approximating training dynamics, especially at initialization (e.g. GraSP). We show that the Path Kernel perspective is not only integral to correctly approximating the behavior of any data-independent pruning approaches but also affects the behavior of a number of other initialization pruning approaches.

The primary goal of this work was not to provide a new method for pruning at initialization that surpasses significantly the performance of those already in existence. Our approach was motivated less by searching for new state-of-the-art results and more by scientific inquiry, hoping to better understand the function and learning dynamics of neural networks through a suitable theoretical lens. We introduced the two Synflow variants to better illustrate and reinforce the understanding gained from the Path Kernel perspective, and we demonstrate the improved performance of these variants under the architectural constraints for which our Path Kernel theory predicts improved performance. We have improved the interpretation of these results in Section 5.2. However, we acknowledge that these performance gains were a bit muddled by our presentation of results. To this end, we have run additional random instantiations for all plots, further narrowing the error bars and aiding interpretation. As requested by Reviewer 3, we have provided a more quantitative analysis of the predictive capacity of the Path Kernel with respect to convergence properties which supplanted our previous visualizations (formerly Figure 1) within the paper. This more quantitative approach has led to additional insights on the predictive capacity of the Path Kernel and its relationship to different pruning approaches and architectures. We have included these insights in Section 5.1.

Importantly, one can compute the Path Kernel at any stage of training, even at convergence. We anticipate the existence of numerous other application areas for which the Path Kernel may offer insight. For example, the Path Kernel provides a data-independent representation of the computational graph of neural networks which could provide serious benefits in the area of model interpretability through representational bias and transfer. We hope that this additional focus on the broader range of applications of this theoretical framework beyond pruning at initialization will make the novelty of this paper much easier to appreciate upon future reading.

---

### Decision · Program_Chairs · 2021-01-07
**Final Decision**

**Decision:**

Reject

**Comment:**

The paper proposes a very interesting decomposition of the neural tangent kernel, which promises
to decouple effects of the parameters and data. The authors illustrate the effects of this decomposition
by considering pruning strategies for initialization.
While the approach looks promising, the current paper is somewhat premature: The only "hard"
theoretical result, Theorem 1, is a direct consequence of the decomposition.  Its consequences for
training discussed in the subsequent paragraph involve quite a few approximations, yet the effects
of these approximations remain unclear. This general, high-level tone is kept when discussing the
initializations.
Finally, the N(0,1)-response to Reviewer 3 worries me.